# Beyond Benchmarks: Spotting Key Topical Sentences While Improving Automated Essay Scoring Performance with Topic-Aware BERT

**Yongchao Wu \*** , **Aron Henriksson, Jalal Nouri, Martin Duneld and Xiu Li**

Department of Computer and System Sciences, Stockholm University, NOD-Huset, Borgarfjordsgatan 12, 16455 Stockholm, Sweden
* Correspondence: yongchao.wu@dsv.su.se

**Abstract:** Automated Essay Scoring (AES) automatically allocates scores to essays at scale and may help teachers reduce the heavy burden during grading activities. Recently, researchers have deployed neural-based AES approaches to improve upon the state-of-the-art AES performance. These neural-based AES methods mainly take student essays as the sole input and focus on learning the relationship between student essays and essay scores through deep neural networks. However, their only product, the predicted holistic score, is far from providing adequate pedagogical information, such as automated writing evaluation (AWE). In this work, we propose Topic-aware BERT, a new method of learning relations among scores, student essays, as well as topical information in essay instructions. Beyond improving the AES benchmark performance, Topic-aware BERT can automatically retrieve key topical sentences in student essays by probing self-attention maps in intermediate layers. We evaluate the performance of Topic-aware BERT of different variants to (i) perform AES and (ii) retrieve key topical sentences using the open dataset Automated Student Assessment Prize and a manually annotated dataset. Our experiments show that Topic-aware BERT achieves a strong AES performance compared with the previous best neural-based AES methods and demonstrates effectiveness in identifying key topical sentences in argumentative essays.

**Keywords:** artificial intelligence; natural language processing; automated essay scoring; automated writing evaluation; BERT

## 1. Introduction

Automated essay scoring (AES), the task of employing natural language processing (NLP) technology to score student essays at scale, plays a vital role in lightening educators' workload [1,2]. Recently, with the raising of massive open online courses (MOOCs), valid and reliable automated assessment tools are vital to test learning outcomes from a large number of learners [3,4]. In addition, during grading activities, the same teacher may allocate different scores for the same essay at different times, and different educators may score differently for the identical essay [5]. AES systems could effectively alleviate this intrarater and inter-rater inconsistency [6]. Two categories of approaches have been investigated to tackle the AES task: feature-based and neural-based approaches. For feature-based approaches, expert knowledge is needed to design linguistic or rubric indices [7–9] reflecting essay grammar, content, and structures, and these manual indices serve as input features for linear regression methods. On the other hand, neural-based approaches automatically learn the features and relations between student essays and their scores through deep learning networks in an end-to-end fashion, eliminating the need for feature engineering and generally outperforming feature-based AES systems [6,10–12]. A vast majority of neural-based AES systems share the same goal of contesting to improve upon the state-of-the-art (SOTA) benchmark performances on a holistic predicted essay score to reflect the general essay quality through designing deep and complex neural network architectures.

Nevertheless, despite their remarkable AES benchmark performance, the neural-based AES systems work like black boxes [7], and their produced holistic scores are far from providing adequate pedagogical information to educators in practice, which raises ethical issues about their algorithms [13]. Researchers have worked on automated writing evaluation (AWE) systems to provide students with formative writing feedback on specific quality dimensions such as coherence, grammar errors, rhetorical moves, and topical development [14–17]. However, these AWE systems are developed based either on separate algorithms divorced from AES systems or on feature-based AES systems targeting specific rubrics with less competitive automated essay scoring results.

It is challenging to design neural-based AES systems that can provide AWE feedback. Unlike feature-based methods whose input features are explicit rubric indices, most neural-based AES systems take the student essay solely as input, neglecting another essential information, the essay instructions. Whereas for a human being's process of composing argumentative essays, they first read and pay attention to the topics in the essay instructions and then conduct argumentative writing *"in a principled way to support a claim using reasons and evidence from multiple sources"* [18]. This human being's reading and writing process inspired us to develop a neural-based AES system that predicts essay scores based on both essay instruction information and student essays. We believe that by feeding essay instruction information, the neural-based AES system could learn prompt-specific knowledge, which would enhance the automated essay scoring performance. In addition, from a pedagogical point of view, student academic success requires the ability to produce a high-quality argument, complete with statements, warrants, and evidence [19,20]. Thus, we aimed to provide AWE feedback on spotting key topical sentences (KTS) from student essays that reflect topical information either by reason or evidence. Together with a predicted score, we believe that spotting key topical sentences in argumentative essays could facilitate the teachers' grading process when judging the quality of essays.

Specifically for our method, we deploy a topical sequence extraction agent to extract topical information from the essay prompt and then we feed both topical information together with student essays to BERT [21] to train the AES system (denoted by Topic-aware BERT). For AWE feedback, we retrieve KTS by ranking the self-attention weights between sentences in student essays and topical information in the trained Topic-aware BERT, motivated by the argument by Clark et al. [22] that self-attention maps in BERT could help understand what neural networks learn about language. To evaluate our method, we use an open dataset to evaluate the AES performance using the official quadratic weighted kappa (QWK) metric. Moreover, we manually create a dataset to evaluate the performance of the model to retrieve KTS. The results show that Topic-aware BERT achieves a competitive AES performance compared to state-of-the-art models and that our KTS-retrieving method is effective when applied to argumentative essays. To the best of our knowledge, this is the first study to train a robust BERT-based AES system with prompt-specific knowledge and among the very few works that develop systems that connect both neural-based AES ad AWE systems. We summarize the contributions of this work below:

1. We successfully link neural AES and AWE by designing a fully automatic multiagent AES + AWE system.
2. We propose Topic-aware BERT and improve AES performance significantly by introducing prompt-specific knowledge.
3. This is the first study to build an AWE system by interpreting self-attention layers in BERT. The experiments show that Topic-aware BERT achieves robust performance in spotting key topical sentences from argumentative essays as AWE feedback, by probing attention weight maps.

This paper is organized as follows. Section 2 lists the related work of automated essay scoring and automated essay evaluation. Section 3 presents the data used in this paper, including the ASAP dataset and a human-annotated KTS dataset. The method formulation of the topical sequence extraction, Topic-aware, and KTS-retrieving approach can be found in Section 3. Experiments settings, baselines and comparison models, evaluation metrics,

as well as empirical results and analysis, are presented in Section 4. Finally, we conclude this paper and discuss future works in Section 6.

## 2. Related Works

### 2.1. Automated Essay Scoring

The research about automated essay scoring systems started with Page's work [23] in the 1960s about building the Project Essay Grader (PEG) system and has remained active since then. In general, the approaches tackling automated essay scoring can be grouped into two categories: feature-based AES and neural-based AES systems. The feature-based AES systems rely on carefully designed compelling features reflecting specific properties of student essays, which require expert knowledge and manual effort. These features cover a variety of aspects of essay properties, including simple surface linguistic features such as essay length [23], and deep linguistic features reflecting grammatical quality [8] or syntactic quality [24], as well as content and structure features such as readability features [25] and coherence features [26]. In order to improve the automated essay scoring performance of feature-based AES, researchers tend to employ a significant number of handcrafted features. For instance, an AES system *e-rater* [27] developed in 2004 was based on ten features, while a recent AES system developed by Kumar et al. [7] in 2021 used 1592 features. On the other hand, neural-based AES systems take student essays directly as inputs and learn the features from training deep neural networks, obviating the need for handcrafted features and generally outperforming feature-based systems. Researchers have employed complex and large neural network architectures to learn essay representations and improve upon the SOTA AES performance. For instance, Taghipour et al. [11] introduced AES methods based on convolutional neural networks and long short-term memory networks (CNN-LSTM). Dong et al. [12] enhanced AES performance by proposing to use global attention and hierarchical networks (CNN-LSTM-Att). Recently, large pretrained language models, such as BERT [21], have shown remarkable effectiveness in different natural language processing tasks and have also been applied in AES tasks. For instance, Yang et al. [6] further improved upon the state-of-the-art performance by modifying the loss function in a BERT-based AES system. The majority of work from both feature-based AES and neural-based AES systems concentrates on predicting a holistic score reflecting the general quality of student essays, competing on benchmark AES performance.

### 2.2. Automated Writing Evaluation

Automated Writing Evaluation systems are designed to provide formative feedback to assist students with essay revisions. The current AWE systems mainly use non-neural features to provide feedback on particular dimensions of essay quality. For instance, Higgins et al. [14] used a support vector machine with identified features from discourse structure to capture essay coherence. Woods et al. [15] provided rubric-specific feedback on the clarity and organization of sentences from a feature-based AES system. Shibani et al. [16] deployed a feature-based discourse analysis framework to provide feedback on rhetorical moves through a web application. Zhang et al. [17] and Madnani et al. [28] built AWE systems that could generate feedback messages related to topic development and topical components, which were based on predefined rubric-based features as well. Constructing useful AWE information from a neural-base AES system is a difficult task because the neural features are automatically and implicitly learned during the end-to-end deep neural network training. Very few studies, such as that of Zhang and Litman [29], have tried to extract AES feedback by exploring information in the intermediate layers of deep neural networks. Specifically, Zhang and Litman interpreted the global attention layer in their LSTM-based coattention neural networks to automatically extract topical components in a specific writing task, response-to-text assessment (RTA), which has a similar spirit to our work. To our best knowledge, there are no systematic studies on probing the self-attention layers in modern large language models (LM), such as BERT, to build AWE systems.

## 3. Data and Methods

In this section, we describe the datasets and methods employed in this work. To evaluate the AES and KTS retrieval performance of Topic-aware BERT, we use an open dataset, Automated Student Assessment Prize, and a manually annotated KTS dataset, which is described in Section 3.1. Then, we present the architecture of Topic-aware BERT and the detailed methods on AES and KTS retrieval tasks in Section 3.2.

### 3.1. Datasets

Researchers have used datasets such as the CLC-FCE datasets [30], the TOEFL11 [31] corpus, and the ASAP dataset to train and evaluate AES systems. As most of the recent best AES models [6,10–12] utilized the ASAP dataset, we also employed the ASAP dataset to conduct a coherent performance comparison. In addition, as the minimum sufficient test size to evaluate an information retrieval system should be 50 [32], we manually constructed a KTS dataset with a size of 51 to evaluate the KTS retrieval performance. The detailed description of the ASAP and the KTS dataset can be found in Sections 3.1.1 and 3.1.2.

### 3.1.1. Automated Student Assessment Prize (ASAP) Dataset

The open dataset Automated Student Assessment Prize (ASAP) dataset ( ASAP dataset: https://www.kaggle.com/c/asap-aes/data, accessed on 10 January 2022) from the Kaggle competition was employed to evaluate the automated essay scoring performance of our proposed approach. In this dataset, there are eight essay sets, and each set contains an independent essay prompt that instructs students to perform essay writing. The ASAP essays have genres. For instance, essays from prompts 1 and 2 are argumentative (ARG) essays; essays from prompts 3 to 6 are response-to-text (RTA); essays from prompts 7 and 8 are narrative (Narrative). According to [7], only essay prompts 1, 2, 7, and 8 truly examined students' writing skills. Considering that we focused on argumentative essays, this study enclosed essays from prompts 1 and 2 in the experiments. We also included narrative essay prompt 7 because we wanted to see how our KTS-retrieving method performed differently on argumentative and narrative essays. Prompt 8 essays were excluded from the experiment because they were too long, and some of the key topical sentences might be truncated by BERT. Some other ASAP dataset statistics, such as essay size and average essay lengths, can be found in Table 1.

**Table 1.** ASAP dataset statistics.

| Prompt | Essay Size | Genre | Avg. Len. |
|--------|-----------|-------|-----------|
| 1 | 1783 | ARG | 350 |
| 2 | 1800 | ARG | 350 |
| 3 | 1726 | RTA | 150 |
| 4 | 1772 | RTA | 150 |
| 5 | 1805 | RTA | 150 |
| 6 | 1800 | RTA | 150 |
| 7 | 1569 | NAR | 250 |
| 8 | 723 | NAR | 650 |

### 3.1.2. Human Annotated KTS Dataset

To evaluate our KTS-retrieving methods, we manually created an evaluation dataset through a clickable web-based annotation tool (The evaluation dataset and annotation tool source code can be found in the Data Availability Statement). Firstly, we recruited three experts with backgrounds in NLP and technology-enhanced learning (TEL) to conduct a prestudy to define the annotation gold standard. During the prestudy, the experts were given 12 randomly selected essays (3 essays from each prompt) and asked to pick five KTS for each essay. From the expert-annotated result, we found that even though some annotations were different, they were related to the different aspects of the same topics. Based on this observation, we decided to take the union of two annotators' annotations as

the gold topical key sentences for each essay. To verify this idea and avoid a motivation and knowledge bias, we employed six PhD students to annotate the same 12 essays following the same procedure as the experts. We first calculated the union scores between the annotations from experts and PhD students. The average union score of annotations among experts and PhD students was 7 and 7.7, respectively, which meant the range of average union score should be between 7.7 and 14.7. Then, we calculated the average union scores (=9) of annotations between the expert and PhD group, which was very close to the minimum score (=7.7), indicating a high overlapping of the annotations. These prestudy results confirmed that using the union of two annotators' annotations was reliable as gold KTS. Overall, we recruited six PhD students to annotate 51 essays randomly selected and in line with the original essay grade distributions following the guidelines from the prestudy to evaluate our KTS-retrieving method.

*3.2. Method*

The overview of our proposed Topic-aware BERT AES system is illustrated in Figure 1. Three essential components comprise the proposed AES system: topical sequence extraction, automated essay scoring, and key topical sentence spotting. Unlike previous methods that take student essays solely as input, the Topic-aware BERT AES system is aware of the essay topics by taking additional input from the topical sequence extraction agent. The Topic-aware BERT can provide two types of enlightening output for teachers: (1) a reliable predicted essay score and (2) spotted key topical sentences from the student essay. In this section, we first explain the topical sequence extraction method from the essay instructions. Then, we describe the model of Topic-aware BERT for automated essay scoring in detail regarding its architecture and mechanism of topic awareness. Last, we present the method of spotting key topical sentences in student essays by interpreting self-attention maps from the transformer layers in Topic-aware BERT.

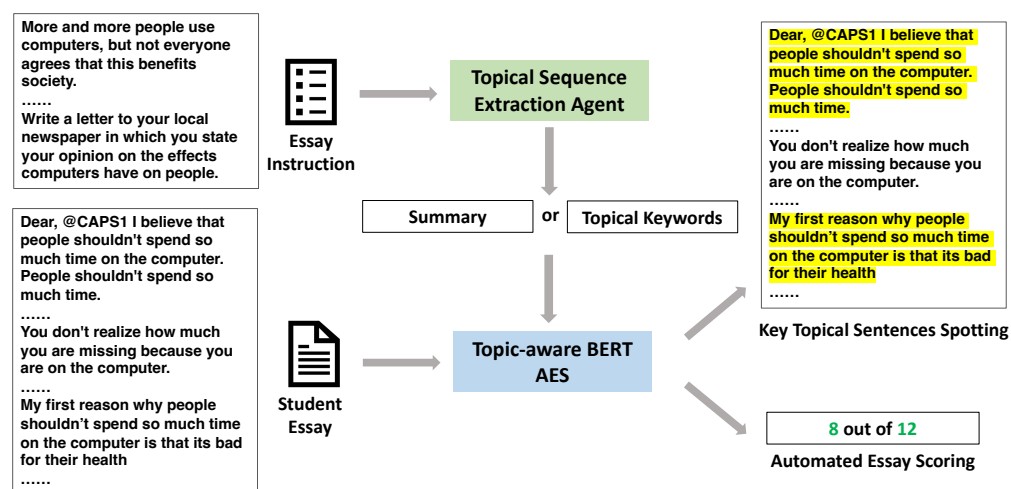

**Figure 1.** The overview of the Topic-aware BERT AES system. A BERT-based AES model is fed in with student essays as well as topical information from the essay instructions as input and can generate a reliably predicted score and spot the key topical sentences.

3.2.1. Topical Sequence Extraction

Considering BERT might truncate student essays if the input sequences exceed the maximum length (512 tokens) it can process [21], we aim to extract short topical sequences from the essay instructions instead of feeding BERT with the whole essay instructions. We explored two variants of short topical sequences: *topical keywords* and *summary*. *Topical keywords* refer to the keywords covering the most topical information from the essay instructions, and *summary* is a short text that conveys a similar meaning to the original long document. For *topical keywords* extraction, we investigated two approaches: manual

topical keyword extraction and automatic key-phrase extraction. For *summary* extraction, we employed two variants of pretrained summarization models to produce single-sentence summaries and multisentence summaries. The strategies to extract short topical sequences are listed as follows:

- Manual topical keyword extraction (denoted by Manual). We recruited three experts with knowledge backgrounds in NLP and TEL to pick the topical keywords from essay prompts 1, 2, and 7. Each expert was asked to construct a list of words as the topical keyword candidates from the essay instructions. Afterwards, we used the NLTK (NLTK Library: https://www.nltk.org/, accessed on 10 January 2022) toolkit to get the lemmatized versions of the topical keyword candidates. Then, we took the intersection of the lemmatized topical keyword candidates from each expert for each prompt as the final topical keywords.
- Automatic key-phrase extraction (denoted by YAKE). We used *YAKE!* [33], the current SOTA unsupervised approach to extract key phrases based on the local statistical features from the single documents, to automate the process of topical keyword extraction. Specifically, we utilized the *YAKE!* python library (YAKE source code: https://github.com/LIAAD/yake, accessed on 10 January 2022) to process essay instructions from prompts 1, 2, and 7 and took the outputs as the topical sequences for each essay prompt.
- Automatic single-sentence summarization (denoted by Xsum). The current SOTA summarization model PEGASUS [34] was deployed as the base model to produce summaries from essay instructions automatically. We used PEGASUS fine-tuned with the XSum [35] dataset for single-sentence summarization. The XSum dataset was constructed with BBC articles covering various subjects together with expert-written single-sentence summaries.
- Automatic multiple-sentence summarization (denoted by CNN). Like single-sentence summarization, PEGASUS also served as the base model for multiple-sentence summarization. Especially, the PEGASUS model, fine-tuned with the CNN/DailyMail [36] dataset, was deployed to generate multiple-sentence summaries from essay instructions. The CNN/DailyMail dataset consists of articles from CNN and Daily Mail newspapers, along with bullet-point summaries.

We summarize the topical sequence extraction method in Table 2. As shown in Table 2, the length of topical sequences extracted with different methods varied from 4 to 42, which might truncate student essays when the concatenation of topical sequence and student essay exceeds 512 tokens. The essay truncation might affect the performance of automated grading and key topical sentence, which is discussed in Section 5. Overall, we propose one manual method (Manual) and three automatic methods (YAKE, Xsum and CNN). The extracted topical sequences via different strategies are listed in Table 3 for illustration. Taking essay instruction prompt 1 as an example, Manual produced a concise list of key topical words *computer, positive effect, concern*, and Xsum generated a short summary *what do you think about the effects computers have on people*; YAKE and CNN extracted longer topical sequences, consisting of key phrases and sentences of summary, respectively.

**Table 2.** Topical sequence extraction method summary.

| Topical Sequence Extraction Methods | Agent Type | Topical Sequence Type | Topical Sequence Length | Essay Truncation? |
|---|---|---|---|---|
| Manual | Manual | Topical keywords | 4 | No |
| YAKE | Automatic | Key phrases, keywords | 32 | Yes |
| Xsum | Automatic | Single-sentence summary | 12 | No |
| CNN | Automatic | Multiple-sentence summary | 42 | Yes |

**Table 3.** Examples of extracted topical sequences via different strategies.

---

**Essay instruction prompt 1**: More and more people use computers, but not everyone agrees that this benefits society. Those who support advances in technology believe that computers have a positive effect on people. Some experts are concerned that people are spending too much time on their computers. Write a letter to your local newspaper in which you state your opinion on the effects computers have on people.

**Manual**: computer, positive effect, concern

**YAKE**: benefits society, people, computers, society, benefits, time, support advances, advances in technology, effects computers, give people, positive effect, teach hand-eye coordination, hand-eye coordination, support, advances, technology, positive, teach hand-eye, ability to learn, learn about faraway

**Xsum**: what do you think about the effects computers have on people

**CNN**: Those who support advances in technology believe that computers have a positive effect on people. Some experts are concerned that people are spending too much time on their computers and less time exercising. Write a letter to your local newspaper in which you state your opinion on the effects computers have on people

---

**Essay instruction prompt 2**: Write a persuasive essay to a newspaper reflecting your vies on censorship in libraries. Do you believe that certain materials, such as books, music, movies, magazines, etc., should be removed from the shelves if they are found offensive?

**Manual**: censorship, library

**YAKE**: children, shelf, Katherine Paterson, Libraries, hope, book, books, work I abhor, Censorship, censorship in libraries, Author, books left, Katherine, Paterson, abhor, remove, work, music, movies, magazines

**Xsum**: what do you think about censorship in libraries

**CNN**: Do you believe certain materials, such as books, music, movies, magazines, should be removed from the shelves if they are found offensive? Support your position with convincing arguments from your own experience, observations, and/or reading.

---

**Essay instruction prompt 7**: A patient person experience difficulties without complaining. Do only one of the following: write a story about a time when you were patient OR write a story about a time when someone you know was patient OR write a story in your own way about patience.

**Manual**: patience, story

**YAKE**: Write, write a story, patient, patience, story, time, understanding and tolerant, patient OR write, Write about patience, tolerant, patient person, difficulties without complaining, patient person experience, understanding, complaining, person experience, experience difficulties, person experience difficulties, person experience

**Xsum**: write a story about a time when you were patient or when someone you know was patient

**CNN**: Do only one of the following: write a story about a time when you were patient. Write a story about a time when someone you know was patient

---

### 3.2.2. Topic-Aware BERT Architecture

The Topic-aware BERT architecture is shown in Figure 2. Let $K$ $(t_{k_1}, t_{k_2}, \cdots, t_{k_n})$ be the topical sequence and $E$ $(t_{e_1}, t_{e_2}, \cdots, t_{e_m})$ be the student essay sequence. To make BERT aware of the topical information, we concatenated the topical sequence $K$ and student essay sequences $E$ with a special $[SEP]$ token.

$$S = concatenate(K, E) \tag{1}$$

The concatenated token sequence, prepended with a $[CLS]$ token, served as the input sequence, denoted by $S = ([CLS], t_{k_1}, t_{k_2}, \cdots, t_{k_n}, [SEP], t_{e_1}, t_{e_2}, \cdots, t_{e_m})$, where $n$ and $m$ are the length of the topical sequence and student essay sequence, respectively. The hidden state from the final layers of Topic-aware BERT, $h_C \in \mathbb{R}^{768}$ for $[CLS]$ (In this work, the $BERT_{base}$ model was deployed with a hidden size of 768 and 12 transformer layers.) represented the entire input sequence $S$ and was used to fine-tune the parameters of Topic-aware BERT end-to-end by adding a logistic regression header for automated essay scoring. Specifically for fine-tuning with automated essay scoring, a feed-forward neural network (FNN) with weight matrix $W_C \in \mathbb{R}^{1 \times 768}$ and bias $b$, together with a *softmax* function, were constructed to map the hidden state $h_C$ to a predicted essay score.

$$\begin{aligned} FNN(h_C) &= Wh_C + b \\ s &= softmax(FNN(h_C)) \end{aligned} \tag{2}$$

A standard regression loss mean squared error (MSE) was computed to learn $h_C$ and $W_C$, as shown in Formula (3), where $r$ are the ground truth scores for essays, and $l$ is the essay sample size.

$$MSE(s, r) = \frac{1}{l} \sum_{i=1}^{l} (s_i - r_i)^2 \tag{3}$$

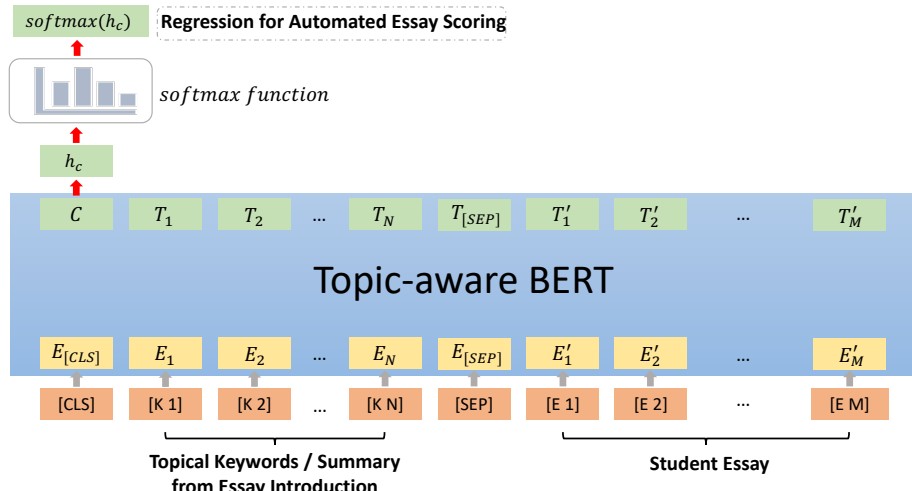

**Figure 2.** Topic-aware BERT architecture. The topical sequences and student essays are concatenated by a special $[SEP]$ token. Topic-aware BERT conducts the AES task by learning the relationship between the predicted essay score and the concatenated input sequence.

### 3.2.3. Using Self-Attention for Retrieval of Key Topical Sentences

BERT consists of multiple layers of transformers [37], and the self-attention mechanism is the essential component. As indicated in Figure 3, the self-attention mechanism is applied to the entire input sequence bidirectionally for each layer of transformers. In Topic-aware BERT, by using self-attention, the sequence of input embeddings $(E_{CLS}, \cdots, E_{SEP}, \cdots, E'_N)$ was mapped to the sequence of output vectors $(Y_{CLS}, Y_1, \cdots, Y'_N)$, where the output vectors consisted of the contextualized information from both topical sequences and student essays. Specifically for the self-attention mechanism, three matrices $Q$, $K$, and $V$, referring to query, key, and value embeddings for each token of the input sequence, were constructed with the learned weight $W_Q$, $W_K$, and $W_V$, respectively. The attention weights between all pairs of tokens were calculated through the softmax of the dot product between the matrix $Q$ and $K$, while the output of the attention head was the weighted sum of $V$, shown in Formula (4).

$$Attention(Q, K, V) = softmax(\frac{Q^T K}{\sqrt{d_k}})V \tag{4}$$

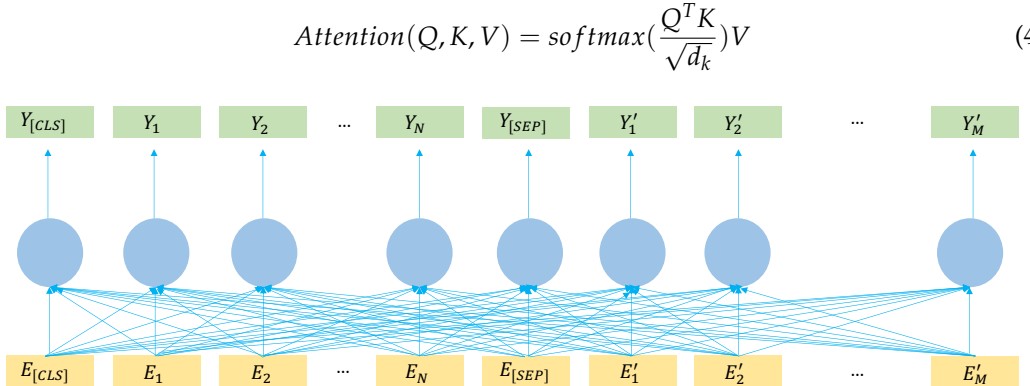

**Figure 3.** Self-attention mechanism in transformer layers.

The attention weight between any two tokens $E_i, E_j$ from the input sequence at layer $l$ was calculated by Formula (5):

$$\alpha_l(E_i, E_j) = softmax(\frac{q_i^T k_j}{\sqrt{d_k}}) \tag{5}$$

where $q_i$ and $k_j$ are the projected vector of $E_i$ and $E_j$ through $Q$ and $K$.

$$\begin{aligned} q_i &= W_Q E_i \\ k_j &= W_K E_j \end{aligned} \tag{6}$$

$\alpha_l(E_i, E_j)$ can be interpreted as the degree of the "importance" of the other token when calculating the new representation of the current token [22]. Inspired by this, we hypothesized that if the average attention scores between the tokens in a sentence from student essays and the tokens from the topical sequences were higher than others, the sentence would be viewed as a more important sentence that affected the representations of the tokens in topical sequences, which could be regarded as the key topical sentences. Thus, we calculated the attention scores between the sentence $s$ and the topical sequence at layer $l$ according to the following formula:

$$Atten\_Sent(s, l) = \sum_{j=1}^{m} \frac{\sum_{i=1}^{n} \alpha_l(E_{s_i}, E_{k_j})}{n} \tag{7}$$

where $n$ and $m$ are the numbers of tokens in the topical sequences and the essay sentences. We ranked the sentences from student essays according to the sentence attention scores in descending order to retrieve key topical sentences as in Algorithm 1. Considering BERT consists of 12 transformer layers, we investigated which layer/layers performed the best by calculating sentence attention scores from each layer ($Atten\_Sent(s, l)$) and all layers as the following.

$$Atten\_Sent(s, all\_layer) = \sum_{l=0}^{11} Atten\_Sent(s, l) \tag{8}$$

---

**Algorithm 1:** Algorithm for Key Topical Sentence Retrieval from Layer $l$

---

**Data:** A student essay $S(s_1, s_2, \cdots, s_n)$ with $n$ sentences
**Result:** Essay sentences sorted by sentence attention weights

1 **for** $i \leftarrow 1$ **to** $n-1$ **do**
2     $v \leftarrow S[i]$;
3     $j \leftarrow i - 1$;
4     **while** $j \geq 1$ & $Atten\_Sent(S[j], l) > Atten\_Sent(v, l)$ **do**
5         $S[j+1] \leftarrow S[j]$ ;                     /* $l \in [0, 1, \cdots, 11, all]$ */
6         $j \leftarrow j - 1$;
7     **end**
8     $S[j+1] \leftarrow v$;
9 **end**

---

## 4. Experiments

This section describes the experiment setup, evaluation metrics, baselines for automated essay scoring, and key topical sentence retrieval.

### 4.1. Setup

The ASAP open dataset was used in this work. According to previous AES works [11,12], we utilized the data partition developed by Taghipour and Ng [11], in which a fivefold cross-validation is adopted. In each fold, essay examples were split into training (60%),

development (20%), and test (20%) datasets. In total, we used 5152 essays from prompts 1, 2, and 7 in our study from each fold to train and validate the AES systems. The Topic-aware BERT AES was implemented based on the $BERT_{base}$ model from Hugging Face (Hugging-face BERT: https://huggingface.co/docs/transformers/model_doc/bert, accessed on 10 January 2022). We fine-tuned the model for ten epochs, with a learning rate of $1 \times 10^{-5}$ and batch size of 10. All experiments were performed on an NVIDIA 3090 GPU. As in previous works, we normalized the essay scores to be in the range between zero and one before training. All the essay scores were mapped back to the original essay scoring scale for measuring quadratic weighted kappa (QWK) scores. We took the best model checkpoints for each essay prompt to evaluate key topical sentence retrieval performance using the mean average precision (MAP) metric.

*4.2. Evaluation Metrics*

In this section, we present the evaluation metrics for automated essay scoring and key topical sentence retrieval.

4.2.1. Evaluation Metrics for Automated Essay Scoring

Quadratic weighted kappa (QWK) is the evaluation metric for AES systems used in previous works [11,12]. To calculate QWK, we first constructed a weighted matrix $W$, where $i$ and $j$ were essay grades assigned by humans and machines, and $N$ was the number of possible grades.

$$W_{(i,j)} = \frac{(i-j)^2}{(N-1)^2} \tag{9}$$

Another two matrices, $O$ (each element stands for the number of essays that receive grade $i$ and $j$) and $E$ (the outer product of histogram vectors of $i$ and $j$), were calculated as well. QWK was calculated as

$$\kappa_{QWK} = 1 - \frac{\sum_{i,j} W_{i,j} O_{i,j}}{\sum_{i,j} W_{i,j} E_{i,j}} \tag{10}$$

4.2.2. Evaluation Metrics for Key Topical Sentence Retrieval

As described in Section 3.2.3, we retrieved KTS through ranking sentences by *Atten_Sent* through each transformer layer. The ranking system of each layer could be regarded as an information retrieval system. Thus, the *mean average precision* (MAP), which has been widely used to evaluate information retrieval systems [32], was also used in this work to evaluate the performance of KTS retrieved from different transformer layer/layers in Topic-aware BERT. The MAP was calculated as :

$$MAP = \frac{1}{Q} \sum_{j=1}^{Q} \frac{1}{m_j} \sum_{k=1}^{n} P@k \times rel@k$$

$$P@k = \frac{\{\text{key topical sentences}\} \cap \{\text{retrieved sentences}\}}{\{\text{retrieved sentences}\}} \tag{11}$$

where $Q$ is the evaluation size and $m_j$ is the number of gold KTS for each essay. $P@k$ is the precision at k, and $rel@k$ refers to a relevance function that equals one if the sentence at rank $k$ is a key topical sentence and equals zero otherwise.

*4.3. Baselines and Comparison Models for Automated Essay Scoring and Key Topical Sentence Retrieval*

To evaluate automated essay scoring performance, we compared our approach against the previous state-of-the-art RNN-based AES approaches, CNN-LSTM [11] and LSTM-CNN-Att [12]. In addition to that, we added non-Topic-aware BERT (denoted by Vanilla BERT) to the baselines to illustrate the effectiveness of topic awareness in automated essay scoring. Since the previous AES models could only do automated essay scoring, we considered

deploying a traditional non-neural approach, termed frequency-inverse document frequency (TF-IDF) [38], and a random selection approach as baselines for the key topical sentence retrieval performance comparison.

The baselines and comparison models for automated essay scoring are itemized below:

- **CNN-LSTM**. CNN-LSTM was proposed by Taghipour and Ng [11], which assembled CNN and LSTM to learn student essay representations.
- **LSTM-CNN-Att**. Dong et al. [12] developed hierarchical CNN-LSTM networks with an attention mechanism to conduct the AES task.
- **Vanilla BERT**. BERT was fine-tuned with student essays as the sole inputs for essay grading, without the awareness of any topical information from essay instructions.
- **Our approach**. Topic-aware BERT was fine-tuned with inputs from both student essays as well as topical sequences extraction by the strategies (Manual, YAKE, Xsum, CNN) mentioned in Section 3.2.1, denoted by **Manual-T BERT**, **YAKE-T BERT**, **Xsum-T BERT**, and **CNN-T BERT**, respectively.

The following summarizes the baselines and comparison models for key topical sentence retrieval:

- **Random**. The sentences from student essays were randomly selected as key topical sentences.
- **TF-IDF**. The sentences from student essays with higher TF-IDF scores were regarded as key topical sentences.
- **Our approach**. The sentences from student essays were ranked by *Atten_Sent* scores from trained checkpoints of **Manual-T BERT**, **YAKE-T BERT**, **Xsum-T BERT**, and **CNN-T BERT**. Sentences with higher *Atten_Sent* scores served as key topical sentences.

## 5. Result and Analysis

This section presents and analyzes the results of automated essay scoring and key topical sentence retrieval experiments.

### 5.1. Automated Essay Scoring Result and Analysis

Table 4 illustrates the QWK scores achieved by baseline models and Topic-aware BERT when conducting automated essay scoring. In general, all baseline models' AES performances were negatively affected by a smaller training data size, as we only conducted experiments with essays from 5152 prompts one, two, and seven, while all prompts consisting of 12,978 essays. For instance, CNN-LSTM achieved a QWK score of 0.821 for essays from prompt one with complete training data, which deteriorated to 0.789 in our experiment. All RNN-based baseline models achieved minor average QWK scores with less training data, where CNN-LSTM's performance fell from 0.772 to 0.760 and that of CNN-LSTM-Att from 0.768 to 0.757. However, there were some exceptions in which some baseline models could benefit from less training data. For example, CNN-LSTM-Att achieved better AES performance (0.825) than when training with full prompts (0.821), and the average QWK of Vanilla BERT improved from 0.767 to 0.774. As topics and grading rubrics differed from prompt to prompt, we suspected that training with fewer prompts of essays resulted in training data with a smaller size but less variety, which could make it easier for non-topic-aware models to learn AES features with training essays of less inconsistency in the topics and grading rubrics.

**Table 4.** QWK scores of different models with essays from prompts one, two, and seven in our experiment. The numbers in the parenthesis indicate the QWK scores reported in the original papers of selected models experimenting with all prompt essays. For example, concerning essays from prompt one, CNN-LSTM achieved 0.789 in our experiments, while the reported QWK in the original paper was 0.821.

|  | Models | Essay Prompt ID | | | Average QWK |
|---|---|---|---|---|---|
|  |  | 1 | 2 | 7 |  |
| Baselines | CNN-LSTM | 0.789 (0.821) | 0.687 (0.688) | 0.805 (0.808) | 0.760 (0.772) |
|  | CNN-LSTM-Att | **0.825** (0.822) | 0.658 (0.682) | 0.788 (0.801) | 0.757 (0.768) |
|  | Vanilla BERT | 0.814 (0.821) | 0.689 (0.678) | 0.820 (0.802) | 0.774 (0.767) |
| Topic-aware BERT | Manual-T BERT | <u>0.822</u> | 0.702 | 0.818 | 0.781 |
|  | YAKE-T BERT | 0.813 | **0.717** | **0.837** | **0.789** |
|  | Xsum-T BERT | <u>0.821</u> | <u>0.710</u> | <u>0.836</u> | **0.789** |
|  | CNN-T BERT | 0.803 | <u>0.714</u> | <u>0.833</u> | <u>0.783</u> |

On the other hand, the results of Topic-aware BERT indicated that the AES systems' performance was significantly improved by introducing topical information from essay instructions. All four variants of Topic-aware BERT showed a robust and competitive AES performance and outperformed all three strong baseline models regarding the average QWK. In fact, Topic-aware BERT achieved the second-best AES performance on essays from prompt one and leads performance over prompts two and seven, beating the baselines trained with either experimented essays or all essays. In particular, the best performance models from Topic-aware BERT outperformed Vanilla BERT, which directly proved that making BERT aware of topical information from essay instructions spurred the AES performance.

Looking at the AES result of the variants of Topic-aware BERT, Manual-T BERT achieved the highest QWK of 0.822 on prompt one, while automatic Topic-BERT (YAKE-T BERT, Xsum-T BERT, CNN-T BERT) gained better performance on prompts two and seven, and similarly for the average QWK. Specifically, Xsum-T BERT performed very close to Manual-T BERT with a QWK score of 0.821. For prompt two, YAKE-T BERT and CNN-T BERT were the best models, acquiring QWK scores of 0.717 and 0.714, respectively. Xsum-T BERT and YAKE-T BERT gained the best performances on prompt three, with QWK scores of 0.836 and 0.837, leading jointly in the average QWK metric with 0.789. This result demonstrated that instead of providing Topic-aware BERT with human-picked topic keywords, feeding it with topical sequences from automatic extraction approaches, such as automated key-phrase extraction and automatic one-sentence or multiple-sentence summarization, could further enhance the AES performance.

*5.2. Key Topical Sentence Retrieval Result and Analysis*

In this section, we present the KTS retrieval empirical result of various variants of Topic-aware BERT (Manual-T BERT, YAKE-T BERT, Xsum-T BERT, and CNN-T BERT) and baselines. Particularly, we analyzed the impact of the topical sequence extraction approaches and different transformer layers of Topic-aware BERT on the KTS retrieval performance with augmentative (prompts one and two) and narrative (prompt seven) essay genres. The KTS retrieval experiment results achieved from different transformers layers of Topic-aware BERT on essays from prompts one, two, and seven are shown in Figures 4–6, respectively.

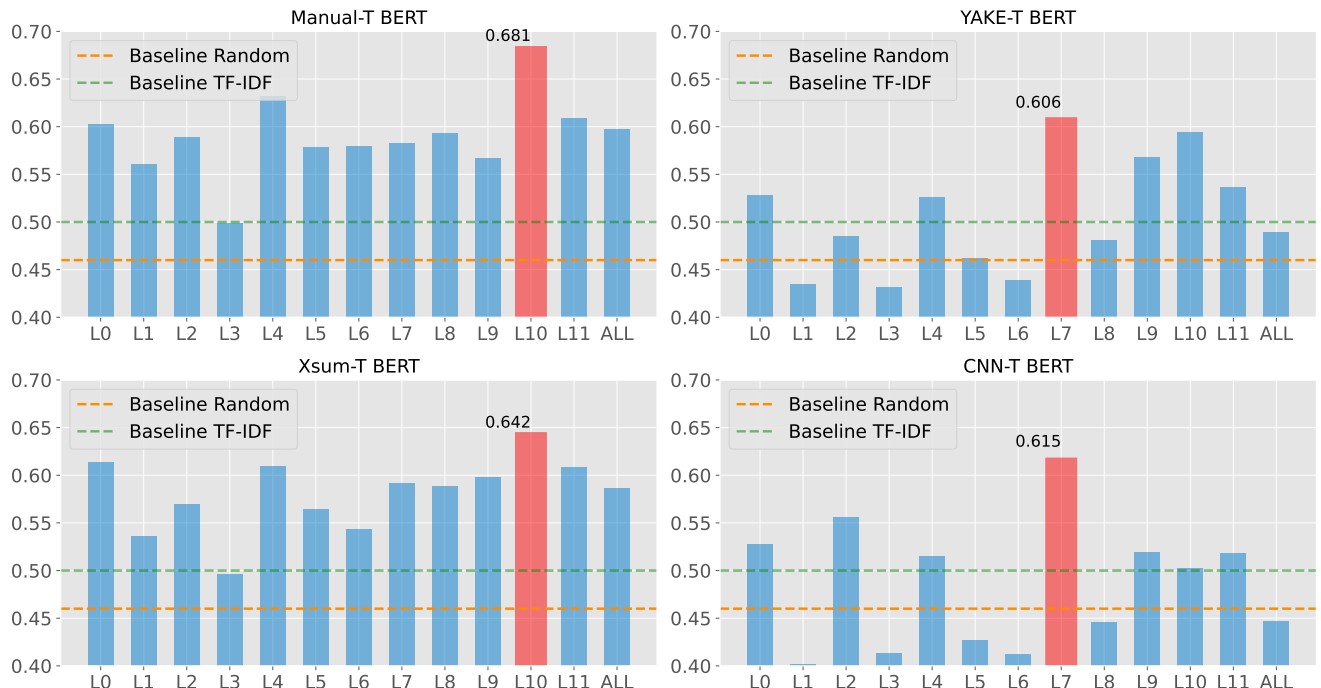

**Figure 4.** KTS performance of Manual-T BERT, YAKE-T BERT, Xsum-T BERT, and CNN-T BERT on essays from prompt 1 (argumentative essays). Baselines of random selection (0.46) and TF-IDF (0.50) are also added. The X-axis indicates different transformer layers $l \in [0, 1, \cdots, 11, all]$. The best-performing transformer layer of each Topic-aware BERT is marked with red, e.g., transformer layer 10 of Xsum-T BERT achieved the highest MAP score of 0.642 among its layers.

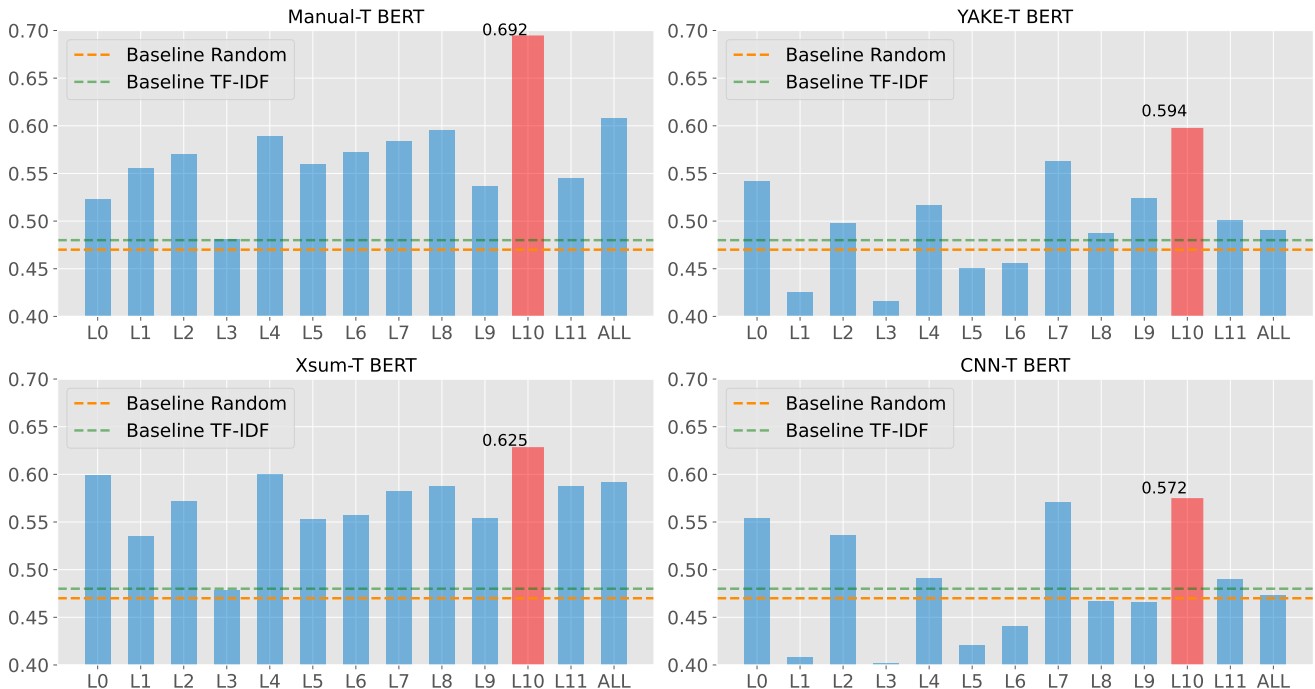

**Figure 5.** KTS performance of Topic-aware BERT variants on essays from prompt 2 (**Argumentative Essays**). Baselines of random selection (0.47) and TF-IDF (0.49) are also added. The X-axis indicates different transformer layers $l \in [0, 1, \cdots, 11, all]$. For YAKE-T BERT and CNN-T BERT, the second-best transformer layers are marked with a green color, e.g., transformer layer 7 of CNN-T BERT gained a MAP score of 0.571, ranking the second among all layers.

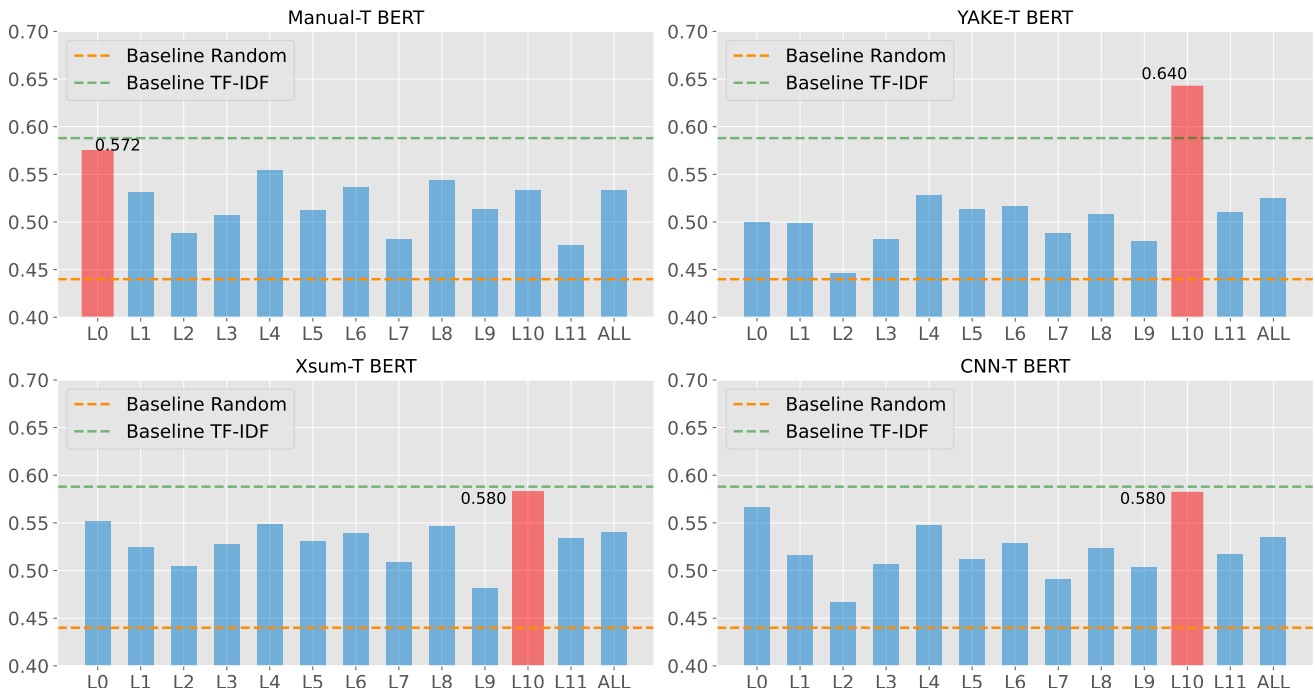

**Figure 6.** KTS performance of Manual-T BERT, YAKE-T BERT, Xsum-T BERT, and CNN-T BERT on essays from prompt 7 (**Narrative Essays**) over transformer layers $l \in [0, 1, \cdots, 11, all]$. Baselines of random selection and TF-IDF are also added.

Figure 4 illustrates the KTS performance of different transformers layers from Manual-T BERT, YAKE-T BERT, Xsum-T BERT, and CNN-T BERT on essays from prompt one with the argumentative essay genre. The random selection and TF-IDF baselines achieved MAP scores of 0.46 and 0.50. The best-performing transformer layers from all four variants of Topic-aware BERT acquired MAP scores over 0.60, defeating baselines with notable margins. In detail, Manual-T BERT performed the best with a MAP score of 0.692, followed by Xsum-T BERT with a MAP score of 0.625. While beating baselines, YAKE-BERT and CNN BERT showed less competitive KTS performance than Manual-T BERT and Xsum-T, achieving MAP scores of 0.606 and 0.615, respectively. This was due to the fact that the topical sequences generated from YAKE and CNN were more extended than from Manual and Xsum. The concatenation of long topical sequences and student essays would exceed the limitation of 512 tokens that BERT could at the most process and would truncate some sentences from the last of student essays. The truncated sentences might contain KTS, stopping Topic-aware BERT from retrieving them. For both Manual-T BERT and Xsum-T, transformer layer 10 outperformed the other layers on KTS retrieval on essays from prompt one, while for YAKE-T BERT and CNN-T BERT, transformers layer 7 performed the best.

A similar pattern of results of KTS performance on essays from prompt two, which were also argumentative, is shown in Figure 5. The best transformer layers of Manual-T BERT, YAKE-T BERT, Xsum-T BERT, and CNN-T BERT outperformed the random selection (MAP 0.47) and TF-IDF (MAP 0.49) baselines. The transformers layer 10 from Manual-T BERT (MAP 0.692) and Xsum-T BERT (MAP 0.625) were the best- and second-best-performing KTS retrieval systems. YAKE-T BERT and CNN-T BERT gained relatively lower MAP scores compared with those of Manual-T BERT and Xsum-T BERT. The transformers layer 10 also served as the best layer in YAKE-T BERT (MAP 0.594) and CNN-T BERT (MAP 0.672) for prompt two essays, while layer 7 acquired a very close KTS retrieval performance with MAP scores of 0.563 and 0.572, respectively. The possibility of truncating essays resulted in the inconsistency of the KTS retrieval performance of YAKE-T BERT and CNN-T BERT between prompts one and two essays.

As shown in Figure 6, the baseline KTS retrieval approach TF-IDF achieved a strong performance with a MAP score of 0.59 on essays from prompt seven with the narrative genre. The transformers layer 10 from YAKE-T BERT was the only KTS retrieval system that outperformed TF-IDF, acquiring a MAP score of 0.640. However, Manual-T BERT, Xsum-T BERT, and CNN-T BERT achieved MAP scores of 0.572, 0.580, and 0.580, failing to surpass TF-IDF. This result indicated that Topic-aware BERT showed a robust KTS retrieval performance when dealing with argumentative essays. Nevertheless, Topic-aware could not effectively retrieve KTS from narrative essays.

### 5.3. Summary of AES and KTS Retrieval Results and Analysis

We summarize the AES and KTS retrieval performance and analysis in Table 5. All variants of Topic-aware BERT outperformed strong AES baselines on the average QWK, demonstrating that with the help of topic sequence extraction, Topic-aware BERT archived a robust and competitive automated essay scoring performance. In addition, the automatic Topic-aware BERT (YAKE-T BERT, Xsum-T BERT, and CNN-T BERT) further improved the AES performance with a thoroughly automatic topic-awareness strategy, which stimulated the scalability of Topic-aware BERT AES systems. Regarding KTS retrieval performance, Manual-T BERT, YAKE-T BERT, Xsum-T BERT, and CNN-T BERT jointly surpass the baseline approaches on essays from prompts one and two, indicating a reliable and effective key topical sentence retrieval performance in argumentative essays. Specifically, as shown in Table 2, Manual-T BERT and Xsum-T had topical sequence lengths of 4 and 12, which were much shorter than those of YAKE-T BERT (32) and CNN-T BERT (42). Correspondingly, Manual-T BERT and Xsum-T demonstrated a more robust KTS retrieval performance than YAKE-T and CNN-T because a longer topical sequence could truncate the student essays and make the KTS at the end of the student essays invisible to Topic-aware BERT. This interesting result revealed an interesting future work to study the relationship between topical sequence lengths and KTS retrieval performance. Looking at the performance of different layers in KTS retrieval, consistently, transformer layer 10 served as the best-performing layer in Manual-T BERT and Xsum-T BERT for argumentative essays. We can recommend layer 10 from Manual-T BERT and Xsum-T to retrieve key topical sentences from student essays. In particular, Xsum-T BERT, as a fully automatic approach, achieved both competitive performances in AES and KTS retrieval, which is promising for a deployment with good scalability. While for the narrative essay from prompt seven, despite the effective automated essay scoring performance, the KTS retrieval competence of Topic-aware BERT exhibited less competitive performance compared with when it processed argumentative essays. We will conduct future work to investigate the relationship between essay genres and KTS retrieval performance. In summary, we conclude that:

- With the awareness of the essay topics, all variants of topic-aware BERT outperformed current best AES baselines on average QWK.
- Automatic Topic-aware BERT further improved the AES performance and indicated a potential for being deployed in practice at scale.
- All variants of topic-aware BERT showed reliable KTS retrieval performance in argumentative essays.
- Topical sequence extraction strategies, such as Xsum, which produced a proper length of topical sequences, could stimulate AES and KTS retrieval performance.

**Table 5.** Summary of AES and KTS retrieval performance and analysis. We record whether or not a Topic-aware BERT variant outperforms the AES and KTS retrieval baselines, as well as their performance ranks, e.g., Xsum-T BERT outperforms AES baselines and achieves the first place among topic-aware models. The best transformers layers on KTS retrieval on different prompts are also recorded, e.g., the 10th layer of YAKE-T BERT obtains the best KTS performance.

| Topic-Aware BERT Variants | | Manual-T BERT | YAKE-T BERT | Xsum-T BERT | CNN-T BERT |
|---|---|---|---|---|---|
| **Outperforms AES baseline on Average QWK?** | | Yes (3rd) | Yes (1st) | Yes (1st) | Yes (2nd) |
| **Outperforms KTS retrieval baseline?** | Prompt 1 | Yes (1st) | Yes (4th) | Yes (2nd) | Yes (3rd) |
| | Prompt 2 | Yes (1st) | Yes (3rd) | Yes (2nd) | Yes (4th) |
| | Prompt 7 | No (3rd) | Yes (1st) | No (2nd) | No (2nd) |
| **Best layer in KTS retrieval** | Prompt 1 | 10 | 7 | 10 | 7 |
| | Prompt 2 | 10 | 10 | 10 | 10 |
| | Prompt 7 | 1 | 10 | 10 | 10 |

## 6. Conclusions and Future Work

In this paper, we proposed Topic-aware BERT to connect automated essay scoring with automated writing evaluation. The experiments illustrated that by feeding both extracted topical sequences and student essays, Topic-aware BERT achieved solid and robust AES performance compared with various previous best AES methods. Moreover, the 10th layer of Topic-aware BERT achieved robust performance in spotting key topical sentences from argumentative essays by probing self-attention scores. With a reliably predicted essay score, the extracted key topical sentences as AWE information will accelerate teachers' grading process, enhance plagiarism control, and improve the transparency of the AES system. In particular, one of the proposed Topic-aware BERT variants, Xsum-T BERT, thoroughly automated the AES and KTS retrieval process and achieved strong performance in both tasks, which had the potential to be widely deployed in practice. We also identified some interesting future work, such as exploring the relationship between the KTS retrieval performance of Topic-aware BERT and essay genres, as well as the topical sequence lengths. As in the previous BERT-based AES system, Topic-aware BERT could only process up to 512 tokens, which might limit the AES and KTS retrieval performance on long essays. Thus, we will investigate models that can process long documents to address this limitation. We will also conduct more experiments with other AES datasets with more genres and varying essay lengths, such as CLC-FCE and TOEFL11, together with larger annotated KTS datasets, to validate the generalization and stability of Topic-aware BERT. In the future, we plan to apply Topic-aware BERT in real classrooms to investigate its scalability and sustainability regarding resource consumption and whether it could benefit teachers during essay-grading activities.

**Author Contributions:** Conceptualization, Y.W. and J.N.; methodology, Y.W., A.H., J.N., and D.M; software, Y.W.; validation, Y.W., J.N., A.H., M.D. and X.L.; formal analysis, Y.W., J.N., A.H., M.D. and X.L.; resources, Y.W. and J.N.; writing—Y.W.; writing—review and editing, Y.W., J.N., A.H., M.D. and X.L.; visualization, Y.W.; supervision, J.N., A.H. and M.D.; project administration, J.N.; funding acquisition, J.N. All authors have read and agreed to the published version of the manuscript.

**Funding:** This research was funded by Swedish Research Council with grant number 2019-05049.

**Institutional Review Board Statement:** Not applicable.

**Informed Consent Statement:** Not applicable.

**Data Availability Statement:** The source code and experimental datasets could be found at the following URL: https://drive.google.com/drive/u/0/folders/1ZxORjHXLvbTbddo1qJesKbdT_SUMpVwj, accessed on 20 December 2022.

**Acknowledgments:** The computations' handling was enabled by resources provided by the Swedish National Infrastructure for Computing (SNIC), partially funded by the Swedish Research Council through grant agreement no. 2018-05973.

**Conflicts of Interest:** The authors declare no conflict of interest.

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
