# Peer review of "Beyond Benchmarks: Spotting Key Topical Sentences While Improving Automated Essay Scoring Performance with Topic-Aware BERT"

_electronics, doi:10.3390/electronics12010150_

Round 1

Reviewer 1 Report

In this paper, the authors propose a Topic-aware BERT that learns relationships between scores, student essays, and topic information in the paper guide, linking automated paper scoring to automated writing evaluation. The topic-aware BERT achieves a competitive AES performance compared to state-of-the-art models. The authors are the first research BERT to build an AWE system by explaining the self-attentive layer in the BERT. The study achieved robust performance in discovering key topic sentences from argumentative essays as AWE feedback. There are no major problems with this article in general, but I still have some suggestions for the author to consider if he is in a position to do so:

1.   In the 3.1.1.Automated Student Assessment Prize (ASAP) dataset, there are fewer essay sets in the genre and other genres can be added as appropriate.

2.   In 3.1.2. Human Annotated KTS dataset, the number of experts and PhDs selected is small and the resulting dataset is not representative.

3.   In 3.2. Method, appropriate introductory content related to topic-aware BERT can be introduced.

4.   In 3.2.2. Topic-aware BERT Architecture, the description of Figure 2 is too brief and a detailed description is recommended.

5.   In 5.3. Summary of AES and KTS Retrieval Result and Analysis, the second paragraph is the summary paragraph, and it is recommended to list them in points in order to make the presentation more readable.

6.   In References, some references are relatively old, and it is desirable to add some recent ones.

Author Response

Dear Reviewer, 

Great thanks for the valuable input!

Please see the attachment for the response. 

If you have any further questions, just kindly let us know. 

Thank you very much!

Reviewer 2 Report

1.Note the indentation of each paragraph, for example, line 250, line 259, etc.

2.Please place the figures and tables reasonably to make the overall paper look neat.

3.Are there any shortcomings to this model? How to remedy this?

Author Response

Dear Reviewer, 

Great thanks for the valuable input!

Please see the attachment for the response. 

If you have any further questions, just let us know. 

Reviewer 3 Report

This paper proposed the topic-aware BERT to complete the automated essay scoring(AES) and key topical sentences(KTS) spotting tasks. If it is applied in real classrooms, I think it will be especially useful for teachers. This paper is organised properly. The authors have conducted a set of experiments and the experiments results are shown clearly. However, I have the following concerns:

1.The authors compare their approach with CNN-LSTM, LSTM-CNN-Att and Vanilla BERT in section 5.1, these comparison methods are neural-based AES systems. I'm not sure whether the feature-based AES and the neural-based AES systems are meaningful for comparison. If so, it would be better to add them. In addition, considering the actual application, whether the running time is also a comparable aspect.

2.The proposed approach takes the topics in the essay instructions into consideration, and the experiments show that it enhances the automated essay scoring performance. But is it possible that these improvements are due to the longer input length, rather than the role of the content(summary or topical keywords) itself? It would be better to design experiments to exclude the impact of input data length, or conduct experiments on more datasets with longer average essay lengths to prove the universality of the algorithm.

Author Response

Dear Reviewer, 

Great thanks for the valuable input!

Please see the attachment for the response. 

If you have any further questions, just let us know. 

Thank you very much!

Reviewer 4 Report

They proposed a new method of learning relations among scores, student essays, as well as topical information in essay instructions.
The manuscript is written in a good and understandable language, and the experiments and results are well presented

Author Response

Dear Reviewer, 

Great thanks for your kind words and encouragement. 

We feel proud and motivated after reading your commendation!

Thank you very much!

Reviewer 5 Report

I found the paper truly interesting. While the technical aspects of the paper are clear to me, I am wondering whether the introduction could give more context on the issues at stake. For example, connecting to online classes, on to the use of Zoom, etc. for classes would be a great point.

Moreover, I am wondering whether you could review some aspects of sociomateriality (e.g. works by Orlikowski or Coron and Porcher 2022) as I believe that some people might not be so keen to use automatic grading. I think that you could discuss potential tensions in conclusion. You migth also refer to the recent debate on ChatGPT.

Author Response

Dear Reviewer, 

Great thanks for the thoughtful comments!

Please see the attachment for the response. 

If you have any further questions, just let us know. 

Thank you very much!
